# Clinical Outcomes of Online Adaptive Magnetic Resonance-Guided Stereotactic Body Radiotherapy of Adrenal Metastases from a Single Institution

**DOI:** 10.3390/cancers16122273

**Published:** 2024-06-19

**Authors:** Philipp Hoegen-Saßmannshausen, Inga Jessen, Carolin Buchele, Fabian Schlüter, Carolin Rippke, Claudia Katharina Renkamp, Fabian Weykamp, Sebastian Regnery, Jakob Liermann, Eva Meixner, Line Hoeltgen, Tanja Eichkorn, Laila König, Jürgen Debus, Sebastian Klüter, Juliane Hörner-Rieber

**Affiliations:** 1Department of Radiation Oncology, Heidelberg University Hospital, 69120 Heidelberg, Germany; 2Heidelberg Institute of Radiation Oncology (HIRO), 69120 Heidelberg, Germany; 3National Center for Tumor Diseases (NCT), 69120 Heidelberg, Germany; 4Clinical Cooperation Unit Radiation Oncology, German Cancer Research Center (DKFZ), 69120 Heidelberg, Germany; 5Department of Radiation Oncology, Heidelberg Ion Beam Therapy Center (HIT), Heidelberg University Hospital, 69120 Heidelberg, Germany; 6German Cancer Consortium (DKTK), Partner Site Heidelberg, 69120 Heidelberg, Germany

**Keywords:** MR-linac, MR-guided radiotherapy, gating, SABR, online adaptation, adaptive radiotherapy, oligometastasis, oligoprogression, NSCLC, melanoma

## Abstract

**Simple Summary:**

Adrenal metastases are frequent in solid malignancies such as lung cancer and melanoma. Recent studies support the use of ablative local therapies in oligometastatic or oligoprogressive patients. Online adaptive MR-guided radiotherapy improves tumor coverage and organ-at-risk sparing compared to non-adaptive radiotherapy. However, clinical data, especially the long-term results of this promising technique, are still limited. We here report the long-term outcomes of a large single-center cohort.

**Abstract:**

(1) Background: Recent publications foster stereotactic body radiotherapy (SBRT) in patients with adrenal oligometastases or oligoprogression. However, local control (LC) after non-adaptive SBRT shows the potential for improvement. Online adaptive MR-guided SBRT (MRgSBRT) improves tumor coverage and organ-at-risk (OAR) sparing. Long-term results of adaptive MRgSBRT are still sparse. (2) Methods: Adaptive MRgSBRT was performed on a 0.35 T MR-Linac. LC, overall survival (OS), progression-free survival (PFS), overall response rate (ORR), and toxicity were assessed. (3) Results: 35 patients with 40 adrenal metastases were analyzed. The median gross tumor volume was 30.6 cc. The most common regimen was 10 fractions at 5 Gy. The median biologically effective dose (BED_10_) was 75.0 Gy. Plan adaptation was performed in 98% of all fractions. The median follow-up was 7.9 months. One local failure occurred after 16.6 months, resulting in estimated LC rates of 100% at one year and 90% at two years. ORR was 67.5%. The median OS was 22.4 months, and the median PFS was 5.1 months. No toxicity > CTCAE grade 2 occurred. (4) Conclusions: LC and ORR after adrenal adaptive MRgSBRT were excellent, even in a cohort with comparably large metastases. A BED_10_ of 75 Gy seems sufficient for improved LC in comparison to non-adaptive SBRT.

## 1. Introduction

Adrenal gland metastases are common in many solid tumors, such as non-small cell and small cell lung cancer, melanoma, and esophageal and renal cancer [1,2,3,4]. In recent years, prospective data have fostered the use of stereotactic ablative body radiotherapy (SBRT and SABR) in patients with oligometastatic and, more recently, oligoprogressive disease [5,6,7,8] as a non-invasive alternative to surgery. Despite metastatic disease, SBRT to oligometastatic or oligoprogressive lesions can prolong progression-free and overall survival, be it in combination with systemic therapy or alone [6,7,8].

Two large multi-center analyses have reported long-term results of non-adaptive adrenal SBRT for 1006 and 260 cases. However, 1-year local control rates of 82.0% and 86.2% [1,9] still leave room for improvement. Even with highly conformal techniques such as robotic SBRT, 1-year local control can be limited to 83.8% [10]. The main factors commonly discussed to limit local control rates are the breathing motion of the adrenal metastasis and the proximity to radiosensitive organs at risk (OAR), as both often restrict target volume dose, especially in larger adrenal metastases. Variable fillings of digestive organs lead to significant interfractional changes [11,12,13]. These challenges can potentially be overcome by the use of online adaptive magnetic resonance-guided SBRT (MRgSBRT). Adaptive MRgSBRT offers superior soft tissue contrast for image guidance and the possibility to account for interfractional changes in tumor position and size as well as OAR position and filling [14,15,16]. Gated dose delivery can further reduce OAR doses in upper abdominal SBRT [17,18]. A particular aspect common in adrenal SBRT is tumor volume change under SBRT, which also necessitates adaptation to ensure target volume coverage [19]. The dosimetric advantage of online adaptation in adrenal SBRT has been demonstrated in several publications [11,12,13,20].

Findings on the dose–response relationship in non-adaptive adrenal SBRT remain inconclusive—most recently, a threshold median biologically effective dose (BED_10_) of 73.2 Gy for the planning target volume (PTV) has been postulated, without an advantage of further dose escalation [1,21,22]. Considering interfractional changes and consecutive dose distributions, if the initial, non-adapted plan was applied to the altered anatomy of a respective fraction, one can assume that the PTV doses actually applied in non-adaptive SBRT are significantly lower than the prescribed doses [11,12,13]. In contrast, one of the key benefits of adaptive MRgSBRT is actually ensuring the dosimetric quality of the initial plan in every single fraction [11]. Thus, the question of which PTV doses really need to be applied to achieve long-term local control can only be answered adequately in consideration of interfractional anatomical changes.

Until recently, long-term clinical results of adrenal SBRT have been limited to non-adaptive radiotherapy techniques without online MR guidance [1,9,23,24]. To the best of our knowledge, only a few very recent publications have assessed long-term clinical outcomes of adrenal MRgSBRT. One cohort was included in two of these works [25,26]. All of these studies had rather small median GTVs, while the median prescribed PTV BED_10_ was at least 100 Gy [25,26,27,28], significantly exceeding the 73.2 Gy threshold postulated by Buergy et al. [21]. Published 1-year local control rates after MRgSBRT are above 90% [25,26,27,28], surpassing the aforementioned results of non-adaptive SBRT. However, it remains questionable if these doses are actually necessary to achieve these outstanding results [26].

The aim of this study was to evaluate the long-term clinical results of adrenal MRgSBRT in a single center and contribute to the rare evidence in that field.

## 2. Materials and Methods

Patients with adrenal metastases treated with online adaptive MR-guided SBRT at a ViewRay MRIdian © 6 megavolt linear accelerator (ViewRay Inc., Denver, CO, USA) between August 2020 and January 2024 were included in the analysis. Patients were enrolled in a prospective single-center registry (institutional ethics board approval S-862/2019), which was designed according to the declaration of Helsinki. All patients were included after written informed consent.

Details of treatment simulation and planning were previously published [11]. Briefly, simulation imaging included a 3D TrueFISP sequence in inspiration breath-hold (axial resolution 1.5 × 1.5 mm^2^ and slice thickness 3 mm) and one or two orthogonal 2D CINE sequences (4–8 frames per second) for gating. For dose calculation, a planning computed tomography (CT) was acquired in the treatment position. Administration of contrast agents was optional.

Gross tumor volumes (GTV) were delineated on the simulation magnetic resonance imaging (MRI), taking into account the co-registered CT scans and diagnostic, contrast-enhanced MRI, as well as positron emission tomography (PET/CT), if available. A clinical target volume (CTV) was created by adding an isotropic margin of 2 mm but respecting borders of non-infiltrated adjacent OAR. The PTV was obtained by adding an isotropic margin of 3 mm. An inhomogeneous dose distribution with a maximum of 125% and a coverage of 95% of the PTV with 100% of the prescribed dose aspired in most of the cases. In some very large metastases, homogeneous dose distribution with a maximum dose of 107% of the prescription was considered appropriate. Institutional constraints for OAR mirrored international guidelines [29]. Prescribed dose and fractionation depended on patient performance status, target volume sizes, OAR proximity, and previous irradiations. In the case of relevant previous irradiation, stricter OAR constraints were chosen individually. Standard institutional constraints for 10 fractions are depicted in Table 1.

Daily online adaptation started with a rigid registration of the MRI of the day and the simulation MRI. OAR and planning CT were co-registered deformably. The GTV was registered without deformation. GTV and OAR were then recontoured within a PTVexpand, 3 cm around the initial PTV (1 cm cranio-caudally) [30]. OAR characterized as parallel or having a mean dose constraint was recontoured completely. Virtual application of the initial plan to the recontoured anatomy of the day led to a predicted plan. The need for plan re-optimization was assessed by the treating physician. Adapted plans were created with the same planning objectives (adjusted, if necessary) and beam parameters as the initial plan. The threshold for automatic real-time gating was 3% and 3 mm.

Toxicity was graded using the Common Terminology Criteria for Adverse Events (CTCAE) 5.0. Local control was graded according to RECIST 1.1 [31]. Local control (LC) was calculated as the time from the first fraction until local failure, defined as the progression of treated lesions according to RECIST 1.1. Overall survival (OS) was calculated as the time from the first fraction until the reported death of any cause. Progression-free survival (PFS) was calculated from the first fraction until tumor progression of any kind or death. PFS and LC were censored at the date of death. Data of patients lost to follow-up were censored at the last follow-up visit. Patients treated bilaterally were considered twice for assessment of LC but only once for the analysis of OS and PFS. In patients treated bilaterally at two different time points, the first series was considered for assessment of OS and PFS. Thus, the total initial number at risk for LC could differ from that for OS and PFS, respectively.

Statistical analysis was carried out with SPSS version 29 (IBM Corporation, Armonk, NY, USA). LC, OS, and PFS were analyzed using non-parametric Kaplan–Meier estimates. Cox regression was used for univariate analyses. *p*-values < 0.05 were considered statistically significant.

## 3. Results

### 3.1. Patient and Treatment Characteristics

Thirty-five patients with 40 adrenal metastases were included in the analysis. Two patients received SBRT to left and right adrenal metastases at two different time points; three patients were treated bilaterally simultaneously. Detailed patient characteristics are shown in Table 2. A percentage of 60% of all patients had non-small cell lung cancer (NSCLC) as a primary tumor. The median Karnofsky performance score (KPS) was 80%. The most common fractionation was 10 fractions with 5 Gy (corresponding to a BED_10_ of 75.0 Gy), applied in 50% of the patients. More than half of all cases were left-sided adrenal metastases.

Of 310 fractions applied in total, the adapted plan was chosen in 305 cases (98%) due to violation of PTV prescription, OAR constraints, or both. An example of stomach constraint violation demanding adaptation is depicted in Figure 1. GTV volumes over the course of MRgRT are shown in Figure 2. Many patients received systemic therapy prior to or after adrenal SBRT, as shown in Appendix A.

### 3.2. Follow-Up

The median follow-up for the overall cohort was 7.9 months. For the patients who were alive at the time of analysis, the median follow-up was 10.7 months. Four patients were lost to follow-up. Of these four patients, one opted for best-supportive care during follow-up, without any further visits or imaging, about one and a half years after adrenal SBRT. One patient was primarily treated at a center several hundred kilometers away and presented at our institution specifically for MRgRT, which was not available closer to her home. After about half a year of follow-up, she did not present at our center again. In two patients, contact was lost two and four months after MRgRT, respectively.

### 3.3. Overall Survival

Twelve patients (34.3%) died during follow-up. The median OS was 22.4 months. Estimated 1-year OS was 67%, and 2-year OS was 48%.

### 3.4. Local Control

Of the 40 adrenal metastases treated, one showed progression 16.6 months after SBRT. This was a patient with NSCLC, treated with five fractions of 10 Gy (BED_10_ = 100 Gy). Two metastases showed minor regrowth after initial shrinkage, with diameters still far below the initial sizes. The objective response rate (ORR) was 67.5%. The 1-year LC was 100%, and the estimated 2-year LC was 90%.

### 3.5. Progression-Free Survival

Twenty-five patients (71.4%) had tumor progression during follow-up. The median PFS was 5.1 months (95% CI: 2.9–7.2 months). The 1-year PFS was 35%, and the 2-year PFS was 6%. Kaplan–Meier curves for OS, LC, and PFS are shown in Figure 3.

In the univariate analysis of various factors potentially influencing OS and PFS (KPS, age, GTV size, primary NSCLC vs. other, stage of oligometastic disease, previous and following systemic therapy, number of metastases and affected organs, sex, previous irradiation, BED_10_, and laterality), only age was a significant predictor of PFS (*p* = 0.014).

### 3.6. Toxicity

No toxicities above CTCAE grade 2 were reported. The most common adverse effects were fatigue, nausea, and anorexia; see Table 3. Of the five patients treated bilaterally, adrenal insufficiency was reported in two patients. One patient did not show impaired adrenal function during follow-up. In two patients, adrenal function during follow-up was not reported.

## 4. Discussion

The present work contributes to improving evidence for online adaptive MRgSBRT of adrenal metastases. To the best of our knowledge, only four publications have reported long-term results of adrenal MRgSBRT so far [25,26,27,28]. With only one local failure and estimated one- and two-year local control rates of 100% and 90% without relevant toxicity, local control was excellent in the present study.

Compared to the other published data on adrenal MRgSBRT, the median GTV in the present cohort (30.6 cc) was significantly larger than the 24.4 cc, 22.6 cc, 21.1 cc, and 22.0 cc reported by others, representing an increase of 25.4%, 35.4%, 45.0%, and 39.1%, respectively [25,26,27,28]. Median GTV was also considerably larger than in a pooled analysis of non-adaptive adrenal SBRT with 22.9 cc (+33.6%) [9]. With larger GTVs, adjacence to critical OAR becomes more likely. This might be a reason why regimens with more fractions were chosen more often in our cohort than in comparable studies. Additionally, the reported frequency of plan adaptation was considerably lower in most studies, with 69% [28], 87.4% [25], and even 26.2% of patients not requiring a single adaptive fraction at all [27]. This also supports the presence of more challenging tumor sizes and tumor/OAR overlaps in the present cohort. During MRgRT, GTV volumes showed variable behavior of growth and shrinkage, as reported previously [19]. The variable number of total fractions in our cohort may prevent the identification of a clear trend. Nevertheless, volumetric variability underlines the need for target volume adaptation.

Previous studies reported grade 3 toxicity rates of 0 to 0.9% with MRgSBRT and 0 to 1.8% in pooled analyses of non-adaptive adrenal SBRT [1,9,25,26,27,28]. The ORR of 67.5% in the current work was very similar to the 62.1%, 65.7%, and 67% shown in comparable studies [25,26,27]. Local control rates of 100% after one year and 90% after two years are also in the range reported by other groups performing MRgSBRT [25,26,27,28], although all these studies applied a median BED_10_ of ≥100 Gy in contrast to the present work with a median BED_10_ of 75 Gy. Our findings support the thesis that a median BED_10_ of approximately 73.2 Gy might be sufficient for adrenal metastases [21]. Compared to pooled analyses of non-adaptive SBRT with reported local control rates of 82–86.2% after one year and 63–75.5% after two years, local control in the present study is significantly higher, although rather large metastases were treated [1,9].

Published OS and PFS rates after online adaptive MRgSBRT or non-adaptive SBRT of adrenal metastases vary significantly. Reported OS rates range from 66 to 91.7% after one year and 42% to more than 70% after two years. PFS rates may vary from 30.9 to 52% (one year) and 16.1% to about 24% (two years). In the present work, 1- and 2-year OS and 1-year PFS lie within this range, while 2-year PFS is lower at an estimated 6%.

Median OS and PFS rates of 22.4 and 5.1 months are also comparable to published data. Notably, the cohort published by Mills et al. has a significantly better PFS and OS. Compared to the present work, these patients had better KPS (KPS 90–100: 57.9% vs. 45.9%) and more patients with genuine oligometastatic disease (83.4% vs. 46%), potentially reflecting fitter patients in a slightly superior stage of the disease.

With the majority of patients being in an oligoprogressive tumor stage, the median PFS can also be compared to the cohort of the randomized CURB study, which reported a median PFS of 7.2 months for the interventional arm and 3.2 months for the control group [8].

In general, the LC, PFS, and OS rates reported before and in the present work mirror the oncological situation of stage IV malignancies. Despite the lack of randomized data, MRgSBRT reproducibly yields superior local control than non-adaptive SBRT. Furthermore, our current results suggest that adaptive MRgSBRT may even allow for the treatment of patients with rather larger and advanced adrenal metastases with excellent local response rates combined with only mild toxicity. As randomized data comparing adaptive and non-adaptive SBRT are unlikely to be published, evidence will probably never reach level 1 or 2.

Limitations of the present study are its retrospective nature, the single-institution character, and the heterogeneity of the cohort regarding primary tumor, previous and following systemic therapy as well as performance status and oncological situation. The median follow-up of 7.9 months was not as long as in some other studies. While the effects of and adjustment to interfractional changes in adrenal SBRT have been clearly quantified and understood [11,12,13], the effects of intrafractional changes, such as peristaltic motion and slow drifts on adaptation results and oncological outcomes, remain to be clarified. In pancreatic SBRT, intrafractional changes may cause further OAR constraint violations despite previous plan adaptation [33]. Therefore, the introduction of OAR planning risk volumes might be one solution to mitigate these effects [34].

Despite larger GTVs, potentially resulting in more GTV/OAR contact and PTV/OAR overlap, the present work showed LC and overall response rates comparable to previous publications on adrenal MRgSBRT, underlining the thesis that a BED_10_ of 75 Gy may be sufficient for excellent local control of adrenal metastases. LC and ORR were considerably superior to the rates reported with non-adaptive SBRT, albeit the median GTV was substantially larger.

## 5. Conclusions

This study demonstrated excellent local control and ORR after adaptive adrenal MRgSBRT, in line with previously published data. Compared to non-adaptive SBRT, adaptive MRgSBRT even enables the treatment of comparably large and advanced adrenal metastases with very good local response rates combined with only mild toxicity. Our results further confirmed that a BED_10_ of 75 Gy might be sufficient for long-term local control of adrenal metastases.

## Figures and Tables

**Figure 1 cancers-16-02273-f001:**
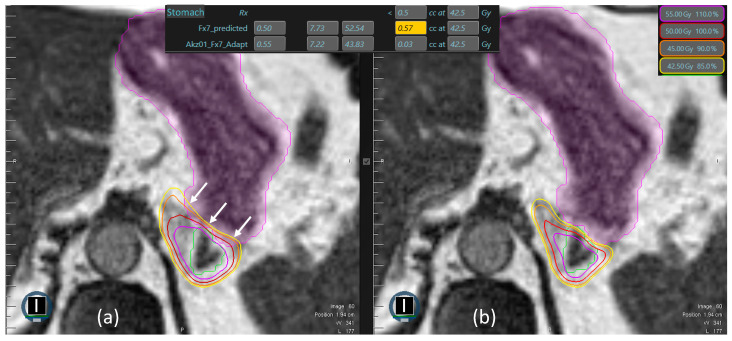
(**a**) Stomach (purple) constraint (0.5 cc < 42.5 Gy) violation in a predicted plan, marked by arrows. Note the excessive maximal dose of 52.54 Gy and the extent of overlap between the stomach and both the 45 Gy (orange) and 50 Gy (red) isodose. After plan adaptation (**b**), the maximum point dose heavily decreased to 43.83 Gy, and the 42.5 Gy isodose conformally respected the boundaries of the stomach. Green line = Gross Tumor Volume (GTV).

**Figure 2 cancers-16-02273-f002:**
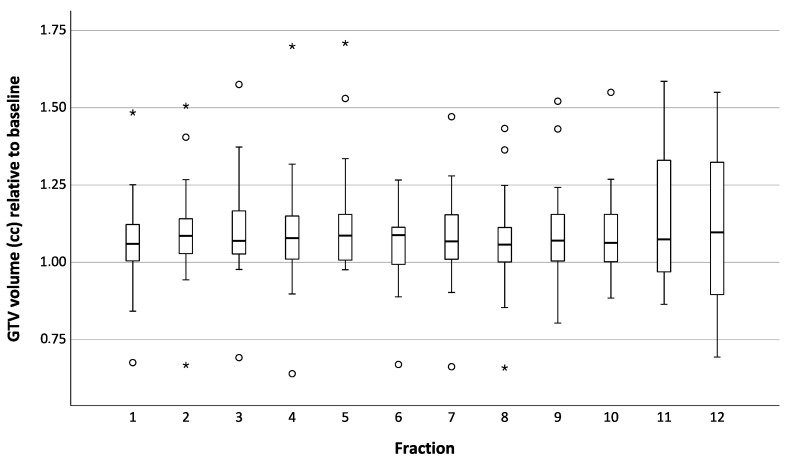
Box plots of GTV volume (cc) over all fractions for all patients. Plots depict median (bar), interquartile range (box), absolute range (whiskers), outliers (dots), and extreme values (stars).

**Figure 3 cancers-16-02273-f003:**
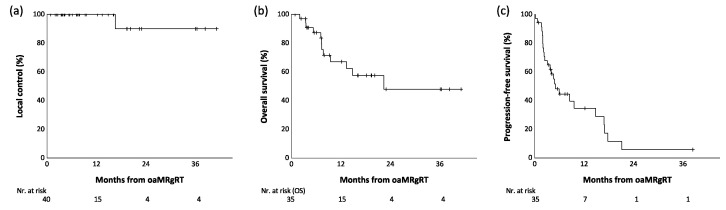
(**a**) Local control, (**b**) overall survival, and (**c**) progression-free survival after online adaptive MR-guided SBRT of adrenal metastases.

**Table 1 cancers-16-02273-t001:** Standard institutional constraints for 10 fractions. D = dose, DXcc = dose to a volume of X cc, ALARA = as low as reasonably achievable.

Organ at Risk	Constraint
Esophagus	D0.5cc < 43.5 Gy
Stomach	D0.5cc < 42.5 Gy
Duodenum and Bowel	D0.5cc < 43.5 Gy
Liver	D mean < 19.5 Gy, 700cc < 32.0 Gy
Kidneys (individual)	D mean < 12 Gy
Spinal cord	D0.1cc < 35.0 Gy
Spleen	ALARA

**Table 2 cancers-16-02273-t002:** Patient characteristics.

Age [y], Median (Range)	63.1 (38.0–81.6)
Sex	
Female	14 (40.0%)
Male	21 (60.0%)
Karnofsky Performance Status	
median (range)	80 (60–90)
90	17 (45.9%)
80	10 (27.0%)
70	7 (18.9%)
60	3 (8.1%)
Primary tumor	
NSCLC	21 (60.0%)
SCLC	3 (8.6%)
Melanoma	5 (14.3%)
Renal cell carcinoma	2 (5.7%)
Esophageal cancer	2 (5.7%)
Pancreatic cancer	1 (2.9%)
Liposarcoma	1 (2.9%)
Laterality	
Left	21 (56.8%)
Right	13 (35.1%)
Bilateral (simultaneous)	3 (8.1%)
Classification of oligometastatic disease according to ESTRO-EORTC consensus definition [32]	
Synchronous oligometastasis	5 (13.5%)
Metachronous oligorecurrence	1 (2.7%)
Induced oligorecurrence	4 (10.8%)
Repeat oligorecurrence	5 (13.5%)
Induced oligoprogression	16 (43.2%)
Repeat oligoprogression	6 (16.2%)
Primary controlled	
Not yet, simultaneous radiotherapy	5
Yes, after surgery	12
Yes, after radio(chemo)therapy	7
At least stable under systemic therapy	11
Imaging prior to SBRT	
CT	9 (24.3%)
CT and PET/CT	20 (54.1%)
CT and MRI	5 (13.5%)
CT, PET/CT, and MRI	3 (8.1%)
Prior adrenal surgery or biopsy	
Surgery	2 (5.0%)
Biopsy	13 (32.5%)
None	25 (62.5%)
Fractionation/BED_10_ [Gy]	
8 × 5.0 Gy/60.0 Gy	2 (5.0%)
12 × 4.0 Gy/67.2 Gy	3 (7.5%)
10 × 5.0 Gy/75.0 Gy	20 (50.0%)
6 × 7.5 Gy/78.8 Gy	2 (5.0%)
4 × 10.0 Gy/80.0 Gy	1 (2.5%)
5 × 10.0 Gy/100.0 Gy	8 (20.0%)
8 × 7.5 Gy/105.0 Gy	4 (10.0%)
GTV size [cc]	
median (range)	30.6 (7.1–292.8)
PTV size [cc]	
median (range)	61.7 (23.7–427.0)
Frequency of adaptation	305/310 (98.0%)
Response per RECIST 1.1	
CR	4 (10.0%)
PR	23 (57.5%)
SD	12 (30.0%)
PD	1 (2.5%)

**Table 3 cancers-16-02273-t003:** Toxicity according to CTCAE V5.0.

	Grade 1	Grade 2
Fatigue	13	6
Nausea	6	1
Emesis	0	0
Anorexia	7	0
Weight loss	1	0
Constipation	1	0
Flatulence	3	0
Other stool alterations	1	0
Adrenal insufficiency	0	2
Pain	3	1
Radiation dermatitis	1	0

## Data Availability

The data presented in this study will be available upon reasonable request from the corresponding author.

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
