# Peer review of "Clinical Outcomes of Online Adaptive Magnetic Resonance-Guided Stereotactic Body Radiotherapy of Adrenal Metastases from a Single Institution"

_cancers, 2024, doi:10.3390/cancers16122273_

Round 1
Reviewer 1 Report
Comments and Suggestions for Authors
Title of the manuscript: Long-term outcomes of online-adaptive MR-guided SBRT of 2 adrenal metastases from a single institution. It is a well written, clinically relevant work, with some important RT technical/dose recommendation messages, so I suggest it for further evaluation. I have only 2 minor notifications/comments:
The loss of 5 out of 35 patients from follow-up is a relatively high rate. Is there any special reason for this?
MRI-based online adaptation could be the answer of several RT technical/anatomical issues, as the authors described it in the manuscript. However, have the authors any data, considering the extent of the decrease in the target volume during RT course?
Author Response
The loss of 5 out of 35 patients from follow-up is a relatively high rate. Is there any special reason for this?
Thank you for this valuable comment. In fact, only four patients were lost to follow-up, but one with bilateral SBRT at two different time points. We have corrected this (l. 162). Of the four patients, one opted for best-supportive care during follow-up, without any further follow-up visits or imaging, about one and a half years after adrenal SBRT. One patient was primarily treated at a center several hundred kilometers away and presented at our institution specifically for online-adaptive MR-guided radiotherapy, which was not available closer to her home. After about half a year of follow-up, she did not present at our center again. In two patients, contact was lost 2 and 4 months after SBRT, respectively.
MRI-based online adaptation could be the answer of several RT technical/anatomical issues, as the authors described it in the manuscript. However, have the authors any data, considering the extent of the decrease in the target volume during RT course?
Thank you for this comment. We have added a respective figure (new Fig. 2) which we also refer to in the discussion. In general, GTV volumes over the course of radiotherapy show large variability, as reported previously by Giraud et al. The variable number of total fractions in our cohort may prevent the identification of a clear trend. Nevertheless, volumetric variability underlines the need for target volume adaptation.
Reviewer 2 Report
Comments and Suggestions for Authors
The present investigation is not original. My conclusion from the Discussion section is that there is no high level evidence that the present results differ significantly from earlier data (refs 25-28). The thesis that a BED10 of 75 Gy may be sufficient for excellent local control of adrenal metastases is not underpinned by the numerical values presented in Table 2 and described on lines 145-46 .
Details:
One.In Table 2, Titel 1 and Title 2 do not explain the contents of the respective columns.
Two. What is the relevance of Table 3?
Author Response
The present investigation is not original. My conclusion from the Discussion section is that there is no high level evidence that the present results differ significantly from earlier data (refs 25-28). The thesis that a BED10 of 75 Gy may be sufficient for excellent local control of adrenal metastases is not underpinned by the numerical values presented in Table 2 and described on lines 145-46.
We appreciate this critical comment. As stated in the manuscript, the literature on clinical outcomes after MR-guided SBRT of adrenal metastases is limited, with only four relevant publications in total to date (refs 25-28). Of these, two are single-center reports (Schneiders et al. and Michalet et al.), and two are multi-center reports (Mills et al., Ugurluer et al.). Notably, the cohort published by Ugurluer et al. also comprised the cohort of Schneiders et al. from Amsterdam. The cohort analyzed in our manuscript is the second largest single-center cohort after the Amsterdam study.
In all published cohorts, the median BED10 was 100 Gy or more. In contrast to experience from pulmonary or hepatic SBRT, the necessity of a BED10 that high for adrenal SBRT is questioned in the literature, e.g. by the large multi-center analysis of Buergy et al. (ref. 9). With a median BED10 of 75.0 Gy, the prescribed doses in our cohort were in the exact range of the dose threshold postulated by Buergy et al. (73.2 Gy). Considering the ALARA principle and the possibility of further radiotherapy treatments at the same anatomic side or level, e.g. of possible contralateral adrenal or spinal metastases, dose should be spared whenever possible. In our opinion, the clinical outcomes achieved in our cohort confirm the dose threshold postulated by Buergy et al.. Interestingly, the one local failure observed in our cohort occurred in a patient with NSCLC who was treated with 5 fractions of 10 Gy (BED10 = 100 Gy). One could argue that higher-level evidence, e.g. from a randomized study, might be necessary to answer this question. However, given the limited availability of MR-linacs and the number of patients needed to achieve statistical proof of equality of dose levels, this project seems challenging, to say the least.
Details:
1. In Table 2, Title 1 and Title 2 do not explain the contents of the respective columns.
Thank you for this correction, we deleted these placeholders.
2. What is the relevance of Table 3?
In our opinion, this table illustrates the multidisciplinary and individual treatment approaches used to treat most of the patients with adrenal metastases in an oligometastatic or oligoprogressive setting. If desired, we could also include this table as a supplementary file instead of placing it in the main manuscript.
Reviewer 3 Report
Comments and Suggestions for Authors
This study provides valuable insights into the outcomes of online-adaptive MRgSBRT for adrenal metastases from a single institution. However, the patient cohort was treated between August 2020 and January 2024, resulting in a relatively shorter median follow-up period (median follow-up of only 7.9 months). Therefore, I would suggest removing 'long-term' from the title. Furthermore, this is a heterogeneous patient cohort, the efficacy of online-adaptive SBRT in controlling adrenal metastases is hindered by various systemic therapy modalities, hence, it is difficult to adjust for these factors in the statistical analyses. Despite these limitations, I believe that the current study still makes contributions to the field and clinical practice.
Author Response
This study provides valuable insights into the outcomes of online-adaptive MRgSBRT for adrenal metastases from a single institution. However, the patient cohort was treated between August 2020 and January 2024, resulting in a relatively shorter median follow-up period (median follow-up of only 7.9 months). Therefore, I would suggest removing 'long-term' from the title.
We thank you for this valuable comment. The title of the manuscript has been adjusted accordingly.
Furthermore, this is a heterogeneous patient cohort, the efficacy of online-adaptive SBRT in controlling adrenal metastases is hindered by various systemic therapy modalities, hence, it is difficult to adjust for these factors in the statistical analyses. Despite these limitations, I believe that the current study still makes contributions to the field and clinical practice.
We absolutely agree to this statement of the reviewer. The heterogeneity of the patient cohort and different systemic treatments is discussed in lines 256-257.
Round 2
Reviewer 2 Report
Comments and Suggestions for Authors
The authors have not sufficiently rebutted my comments, namely lack of originality and low levels of evidence. My advice: reject.
